# MDTree: A Masked Dynamic Autoregressive Model for Phylogenetic Inference

## Abstract

Phylogenetic tree inference, crucial for understanding species evolution, presents challenges in jointly optimizing continuous branch lengths and discrete tree topologies. Traditional Markov Chain Monte Carlo methods, though widely adopted, suffer from slow convergence and high computational costs. Deep learning methods have introduced more scalable solutions but still face limitations. Bayesian generative models struggle with computational complexity, autoregressive models are constrained by predefined species orders, and generative flow networks still fail to fully leverage evolutionary signals from genomic sequences. In this paper, we introduce MDTree, a novel framework that redefines phylogenetic tree generation from the perspective of dynamically learning node orders based on biological priors embedded in genomic sequences. By leveraging a Dynamic Ordering Network to learn evolutionarily meaningful node orders, MDTree autoregressively positions nodes to construct biologically coherent trees. To further push its limits, we propose a dynamic masking mechanism that accelerates tree generation through parallel node processing. Extensive experiments show that MDTree outperforms existing methods on standard phylogenetic benchmarks, offering biologically interpretable and computationally efficient solutions for tree generation.

## 1 Introduction

Phylogenetic trees serve as essential tools for deciphering evolutionary relationships among species, enabling researchers to trace lineages from common ancestors to present-day organisms through DNA or protein sequences (Brocchieri, 2001; Munjal et al., 2019). Their applications permeate diverse fields such as taxonomy, evolutionary biology, and medicine, where they unlock pivotal insights into species origins, decode the genetic blueprints behind biodiversity, and map the intricate evolutionary pathways of pathogens and cancer cells (Hugenholtz et al., 2021; Hosner et al., 2016), driving transformative discoveries in adaptation and survival mechanisms.

Traditional statistical approaches like Maximum Likelihood Estimation (Izquierdo-Carrasco et al., 2011; Huelsenbeck et al., 2001) and Bayesian Inference (Zhang & Matsen IV, 2018a; Wang et al., 2020) via Markov Chain Monte Carlo (MCMC) have long been the cornerstone of phylogenetic inference. However, as species numbers grow, these methods face significant computational hurdles due to the exponential growth in possible tree topologies—$(2N - 5)!!$ for unrooted bifurcating trees—and the complexity of optimizing both continuous branch lengths and discrete tree structures.

Leveraging deep learning, breakthroughs in phylogenetic inference have burst onto the scene, addressing long-standing computational challenges in the field (Nesterenko et al., 2022; Smith & Hahn, 2023; Tang et al., 2024). Research efforts primarily follow two main directions: representation learning on known tree structures and generative models. The former, exemplified by VBPI-GNN (Zhang, 2023), optimizes performance based on predefined topologies but struggles when the topology is unknown and both topology and branch lengths must be inferred. These methods also underutilize evolutionary information from biological sequences, impacting accuracy and flexibility (Penny, 2004). On the other hand, generative models, which infer tree structures directly from data, can be further divided into three types: Bayesian generative models (e.g., Geophy (Mimori & Hamada, 2024)) leverage probabilistic frameworks to capture uncertainty but are computationally intensive; autoregressive models (e.g., ARTree (Xie & Zhang, 2024)) sequentially add nodes, offering flexibility yet relying on predefined orders that overlook true evolutionary relationships, while their stepwise nature leads

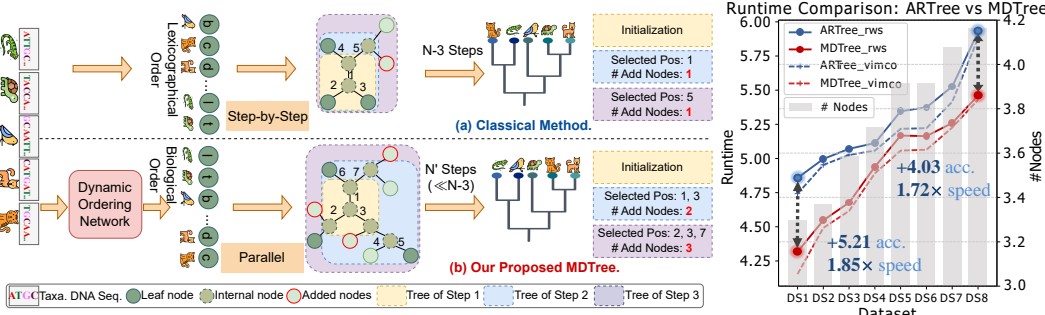

Figure 1: **Comparison of MDTree with classical method for phylogenetic tree construction. Lef:** **(a)** shows the classical autoregressive method, where nodes are added step-by-step in lexicographical order, with one node added by step. **(b)** shows our method, which employs a Dynamic Ordering Network to determine biologically meaningful orders, enabling multiple nodes to be added in parallel at each step. Colored boxes (yellow, blue, purple) indicate the tree structures generated in the first three steps, showing that MDTree covers a broader portion of trees per step, accelerating the generation process compared to the classical method. **Right:** Log-scale comparison of runtime (seconds) and node count between MDTree and ARTree across eight benchmarks with two optimization techniques.

to inefficiency for large datasets (Razavi et al., 2019). Lastly, Generative Flow Networks (GFNs) (e.g., PhyloGFN (Zhou et al., 2023a)) provide greater flexibility by exploring multimodal posterior distributions but still struggle to fully integrate evolutionary signals, impacting the accuracy of inferred trees. Therefore, none of the previous methods achieved these goals simultaneously.

To overcome these limitations, we focus on a core question: *how can biological priors effectively guide node addition to improve phylogenetic inference accuracy?* As shown in Fig. 1, classical autoregressive methods (Fig. 1a) rely on fixed orders (e.g., lexicographical), overlooking evolutionary relationships and may produce inaccurate trees (Hayes et al., 2024). Our method (Fig. 1b) learns evolutionarily meaningful node orders, ensuring species like reptiles, birds, and mammals are added in line with their ancestry. This improves the accuracy and biological relevance of generated trees by prioritizing species with closer common ancestors. Specifically, we redefine phylogenetic tree generation as a Dynamic Autoregressive Tree Generation (DART) task, where genomic sequences serve as input for autoregressive tree construction. Unlike traditional methods that depend on prede-fined orders, DART dynamically optimizes node order and insertion positions. Then, we propose the Masked Dynamic Autoregressive Model (MDTree). MDTree utilizes a Dynamic Ordering Network (DON) to learn biologically informed orders directly from sequence data via an absorbing diffusion model (Bond-Taylor et al., 2021), mitigating the limitations of fixed or random orders. By combining the strengths of Graph Neural Networks and Language Models (LMs), MDTree captures intricate genomic relationships while modeling complex tree structures. A Dynamic Masking Mechanism enables parallel node processing, improving efficiency. Lastly, we employ a dual-pass tree traversal strategy for branch length estimation and use the LAX model (Grathwohl et al., 2017) to reduce variance in discrete sampling for stabilizing optimization and enhancing convergence. Experiments on phylogenetic benchmarks show that MDTree outperforms existing methods in accuracy and efficiency. Empirical analysis of Angiosperms353 (Zuntini et al., 2024) further demonstrates its ability to recover evolutionary lineages, including Rosaceae and Moraceae, suggesting broader biological applications. In summary, our contributions are summarized as follows:

- **Redefinition of Phylogenetic Tree Generation**: From a fresh perspective, we redefine the phylogenetic tree generation task as DART, which dynamically learns node order and insertion positions based on genomic sequence data for more accurate evolutionary relationships.

- **Innovative Methodology**: We propose MDTree, which integrates DON for biologically in-formed node orders, integrates genomic LMs with dual-traversal techniques for precise tree generation, and is coupled with a dynamic masking mechanism for efficient parallel processing.

- **Experimental Results**: Comprehensive experiments validate that MDTree achieves SOTA performance. Visualizations from real-world Angiosperm datasets further confirm the biological relevance and interpretability of the generated trees.

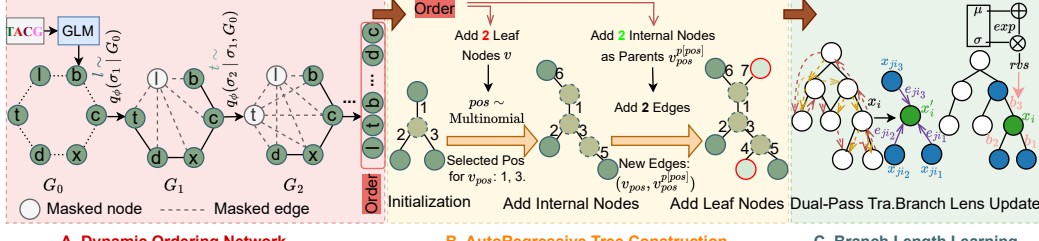

Figure 2: **Framework of MDTree for dynamic autoregressive tree generation. A. Dynamic Ordering Network** module utilizes a pre-trained enomic LM to extract embeddings from sequences $Y$, guiding nodes into absorbing states in an autoregressive manner as determined by DON $q_\phi(\sigma|G)$. **B. Autoregressive Tree Construction** module employs a parallel strategy to add multiple leaf and internal nodes simultaneously at specified positions based on the order provided by DON. **C. Branch Length Learning** module optimizes branch lengths through a dual-pass traversal.

## 2 RELATED WORKS

Phylogenetic inference methods are generally classified into traditional and deep learning-based approaches, further divided into graph structure generation and graph representation models. For a detailed background, refer to Appendix A, and for related work, refer to Appendix B.

**Traditional Methods** rely on predefined evolutionary models and statistical inference. *Graph Structure Generation Models:* MrBayes (Ronquist et al., 2012) utilizes Bayesian inference to generate trees but struggles with high-dimensional combinatorial spaces, requiring large sample sizes for accuracy. VaiPhy (Koptagel et al., 2022) combines SLANTIS sampling strategy (Diaconis, 2019) with biological models (e.g., JC model (Munro, 2012)) to estimate branch lengths and generate accurate tree structures. *Graph Structure Representation Models:* SBN (Zhang & Matsen IV, 2018a) models the probability distribution of tree topologies from existing trees, focusing on subsplit relationships without directly estimating branch lengths. VBPI (Zhang & Matsen IV, 2018b) extends SBNs to estimate posterior distributions and optimize branch lengths through variational inference.

**Deep Learning-based Methods** offer more flexible and scalable solutions. *Graph Structure Generation Models:* (1) Bayesian Generative Models like GeoPhy (Mimori & Hamada, 2024) learn latent tree representations to generate diverse topologies. (2) Autoregressive Models such as ARTree (Xie & Zhang, 2024) sequentially generate trees, well-suited for hierarchical data. (3) Generative Flow Networks like PhyloGFN (Zhou et al., 2023a) optimize tree generation paths using Markov decision processes. *Graph Structure Representation Models:* VBPI-GNN (Zhang, 2023) combines SBNs with variational inference to optimize topology and branch lengths.

## 3 METHODS

**Formulation.** Given $N$ species sequences $\mathcal{S} = \{s_i\}_{i=1}^N$ and a corresponding set of genomic representations $G = \{g_i\}_{i=1}^N$, we can model the phylogenetic tree as a graph $\mathcal{G}_T = (\mathcal{V}_T, \mathcal{E}_T)$, where each node $v_i \in \mathcal{V}_T$ represents a species $s_i \in \mathcal{S}$ and each edge $e_{ij} \in \mathcal{E}_T$ reflects the evolutionary relationship between species $i$ and $j$. Our goal is to autoregressively generate the unrooted binary tree topology $\tau$ and its branch lengths $B_\tau$, representing evolutionary distances. We reformulate the phylogenetic inference problem as the DART task, aiming to learn a mapping $\mathcal{F} : \mathcal{S} \to (\tau, B_\tau)$ that adjusts node orders and insertion positions based on genomic representations, overcoming the limitations of fixed node orders (Xie & Zhang, 2024).

**Framework.** To optimize node addition orders in phylogenetic tree generation, we propose MDTree as shown in Fig. 2, leveraging biological priors embedded in genomic sequences. A Genomic LM (e.g., DNABERT2 (Zhou et al., 2023b)) encodes each sequence $s_i$ into $g_i$, which serve as inputs for the DON $q(\cdot)$ to learn evolutionarily meaningful orders using an absorbing diffusion model (Austin et al., 2021). The optimized order guides the autoregressive generation process, while a dynamic masking mechanism facilitates parallel processing. Finally, branch lengths are refined via a dual-pass traversal, ensuring biological coherence and accuracy in the generated tree to serve multiple downstream tasks.

## 3.1 DON FOR LEARNING BIOLOGICALLY INFORMED NODE ORDERS WITH GENOMIC PRIORS

As discussed, the biologically relevant node addition order is crucial for phylogenetic inference, as species with closer ancestry should be prioritized (Penny, 2004; Gregory, 2008). Despite evidence of robustness across different taxa orders (Xie & Zhang, 2024), the influence of node orders on phylogenetic accuracy has not been thoroughly examined, which is the focus of our work. To this end, we propose DON to learn optimal node orders by leveraging both genomic information and evolutionary relationships between species. The process begins by using genomic LMs (e.g., DNABERT2 (Zhou et al., 2023b)) to encode each species sequence $s_i$ into representations $g_i$, capturing biologically meaningful signals that reflect evolutionary proximity, i.e., species with closer ancestry will have more similar genomic features (Franceschi et al., 2019; Delsuc et al., 2005). These representations serve as inputs to a Relational Graph Convolutional Network (RGCN) (Schlichtkrull et al., 2018) to update node features $h^t$:

$$h_i^t = \text{RGCN}(g_i + \text{PE}(t), e_{ij}), \tag{1}$$

where $\text{PE}(t)$ is the positional encoding for time step $t$. This ensures that local genomic signals and global evolutionary relationships are captured in $h^t$, aligned with the biological prior that closely related species should be placed closer in the tree. Subsequently, a node is selected to transition into an absorbing state $m$, i.e., it is set to a masked value $m = N + 1$, leading to a masked graph $G_t$ where associated edges are also masked. The transition probabilities are defined by the matrix $Q_t$:

$$[Q_t]_{ij} = \begin{cases} 1 & \text{if} \quad i = j = m \\ 1 - \beta_t & \text{if} \quad i = j \neq m \\ \beta_t & \text{if} \quad j = m \quad \text{and} \quad i \neq m \\ 0 & \text{otherwise,} \end{cases} \tag{2}$$

where $\beta_t$ increases monotonically from $10^{-7}$ to $2 \times 10^{-3}$ as the time step $t$ progresses, ensuring all nodes are eventually absorbed. To preserve graph structure, each absorbed node connects to all remaining nodes, maintaining continuity despite masking. This process continues until the entire graph is absorbed. Then, the cumulative transition matrix $\bar{Q}_t = \prod_{i=1}^{t} Q_i$ predicts the node order:

$$q(h^t|h^0, h^{(<t)}) = \text{Cat}(\frac{h^t Q_t^\top \odot h^0 \bar{Q}_{t-1}^\top}{h^0 \bar{Q}_t h^{t\top}}), \tag{3}$$

where Cat is a categorical distribution. At each step, the node $i^*$ with the highest transition probability is selected and added to the tree structure: $G_{t+1} = G_t \cup \{i^*\}$. This process repeats until all nodes are incorporated, with the final node order denoted as Rank, where $\text{Rank}_i$ is the rank of node $i$.

## 3.2 AUTOREGRESSIVE TREE CONSTRUCTION WITH DYNAMIC NODE INSERTION

Once the node addition order is determined, the next step is to decide the optimal insertion positions of the selected nodes. The nodes to be inserted at each step are dynamically selected based on a mask rate modulated by a cosine function. As the mask rate decreases, the number of nodes, $U$, available for insertion increases, enabling parallel processing. These nodes are passed through a Multi-Head Attention (MHA) block (Vaswani, 2017) with 4 attention heads, generating $\mathbf{r}_i = \text{MHA}(Q, h_i, h_i)$, and the query matrix $Q \in \mathbb{R}^{(N-3) \times d}$ is initialized as an identity matrix, with $d = 100$. The probability $\text{L}_i$ for each node's potential insertion position is:

$$\text{L}_i = \text{softmax}(\text{MLP}(\text{Concat}(r_i, \text{MAX}(r_i, r_i^p)) + \text{PE}(t))), \tag{4}$$

where $\text{MAX}(\cdot)$ is the element-wise maximum between features of node $r_i$ and its parent $r_i^p$. To ensure biologically coherent insertion positions, the probabilities are adjusted according to the learned order, prioritizing nodes with higher ranks (closer ancestry) for earlier insertion:

$$\text{L}_{\text{adjusted},i} = \text{L}_i + \alpha \times (N - \text{Rank}_i), \tag{5}$$

where $\alpha$ modulates the influence of the node's priority. The final positions for new nodes are sampled via multinomial sampling:

$$\text{pos} = \text{Multinomial}(\text{softmax}(\text{L}_{\text{adjusted}})). \tag{6}$$

Furthermore, internal node features are computed as the average of neighboring nodes as shown in Fig. 1. After determining the insertion positions, the tree is updated by connecting new nodes to their

---

**Algorithm 1** Phylogenetic Tree Generation using MDTree

---

1: **Input:** Gene sequences $s_i$.
2: Initialize node order using DON.
3: **for** $t = 1$ to $T$ **do** ▷ Iterate for dynamically determined steps
4:     Compute features $r_i$ for $U$ unmasked nodes on masked graph $G_t$ via MHA block.
5:     Update cumulative transition matrix $\bar{Q}_t$ to predict node absorption order.
6:     Add the highest priority node $i^*$ to graph $G_{t+1}$ based on $\text{Rank}_i$.
7:     Adjust node position probabilities $\text{L}_{\text{adjusted}} = \text{L} + \alpha \times (N - \text{Rank})$ (Eq. 5).
8:     Sample final positions pos $\leftarrow$ Multinomial(softmax($\text{L}_{\text{adjusted}}$)) (Eq. 6).
9:     Update tree structure by adding new nodes and edges (Eq. 7).
10:     Update node features $x_i \leftarrow c_i \cdot r_i^{p[i]} + f_i$ using Dual-Pass Traversal (Eq. 8).
11:     Sample branch lengths using reparameterization and compute log-probability $\log q$.
12: **end for**
13: Minimize $\mathcal{L}$ (Eq. 16).
14: **Return:** Final tree $\tau$ and cumulative log-probability $\log p(\tau)$ of branch lengths.

---

parents: $\text{E}' = \text{Concat}(\text{E}, E_{new})$, where $E_{\text{new}}$ represents the edge between newly inserted node $v_{\text{pos}}$ and its parent $v_{\text{pos}}^p$, ensuring a valid range:

$$E_{new} = \begin{cases} (v_{pos}, v_{pos}^{p[pos]}), (v_{pos}^{p[pos]}, v_{pos}) & \text{if } v_{pos}, v_{pos}^{p[pos]} < \text{N} \\ \text{N} + 1 & \text{otherwise.} \end{cases} \tag{7}$$

### 3.3 DUAL-PASS TRAVERSAL FOR BRANCH LENGTH LEARNING

We employ a linear-time dual-pass traversal to estimate branch lengths. In the postorder traversal, features are aggregated from leaves to the root with a scaling factor $c_i = (1 + K - \sum c_j)^{-1}, j \in ch[i]$ adjusting contributions from child nodes. Initially, $c_i = 0$ and $f_i = r_i$. The preorder traversal incorporates parent information from root to leaves:

$$x_i = c_i \cdot r_i^{p[i]} + f_i, \quad f_i = c_i \cdot f_j + r_i, \tag{8}$$

where $K = 3$ corresponds to the binary tree properties. To further enhance node features, we apply a Dynamic Graph Convolutional Network (DGCNN) (Manessi et al., 2020), transforming $x_i^L \in \mathbb{R}^{768}$ into $x_i^{L+1} \in \mathbb{R}^{100}$, with MAX pooling to highlight the most important features. DGCNN outputs are then used to parameterize branch length distributions via an MLP network:

$$z_i = \text{MLP}(x_i^{L+1}), \quad z_i' = \text{MAX}(z_i, z_i^{p[i]}). \tag{9}$$

The mean and log-variance of branch lengths $b$ are derived as: $\mu_b, \log(\sigma_b^2) = MLP(z_i')$. Branch lengths are sampled using the reparameterization trick (Kingma & Welling, 2013) as $b = \exp(\mu_b + \exp(\sigma_b) \cdot \text{rvs})$, where rvs $\sim \mathcal{N}(0, I)$ represents samples from a standard normal distribution. The corresponding log probability is: $\log q_b = \sum_i (-\frac{1}{2}\log(2\pi) - \frac{1}{2}\log(\sigma_{bi}^2) - \frac{(b_i - \mu_{bi})^2}{2\sigma_{bi}^2})$.

### 3.4 MDTREE INFERENCE FOR TREE TOPOLOGY AND BRANCH LENGTH ESTIMATION

As mentioned, phylogenetic inference involves optimizing both continuous branch lengths and discrete tree topologies. To assess MDTree's performance, we design two tasks following (Zhang, 2023; Xie & Zhang, 2024): Tree Topology Density Estimation (TDE) optimizes the tree topology by maximizing the marginal log-likelihood (MLL), while Variational Bayesian Phylogenetic Inference (VBPI) approximates the joint posterior distribution of tree topology and branch lengths using VI via the Evidence Lower Bound (ELBO).

**Task1: Tree Topology Estimation for TDE.** TDE assesses the model's ability to estimate tree topologies by maximizing the log-likelihood of known structures from MrBayes. To further validate model performance, we compare the Kullback-Leibler (KL) divergence between the model-generated tree topology distribution $q_\theta(\tau)$ and the true posterior $p(\tau)$.

**Task2: Joint Optimization of Topology and Branch Lengths for VBPI.** VBPI extends TDE by jointly optimizing tree topology and branch lengths using VI. Unlike TDE, VBPI does not rely

on known tree structures; instead, it approximates the joint posterior distribution by maximizing ELBO, generating tree structures that align with the input gene sequences. Before constructing the phylogenetic tree, the node order $\sigma$ is determined through DON, with the loss $\mathcal{L}_{\text{DON}}$:

$$\mathcal{L}_{\text{DON}} = -\sum_{t=1}^{T} \log q_\sigma(\sigma_t | \mathcal{G}_0, \sigma_{(<t)}). \tag{10}$$

In VI for discrete and high-dimensional parameter spaces, gradient estimation often suffers from high variance and instability during optimization. To address these, we employ two optimization techniques: RWS (Bornschein & Bengio, 2014) and VIMCO (Mnih & Rezende, 2016). Both involve sampling tree topology $\tau$ and branch lengths $B_\tau$ from the variational distribution, followed by computing the log-likelihood $\log p(y|\tau, B_\tau)$ and log prior $\log p(\tau, B_\tau)$. The joint log-probability is then calculated as $\log p_{\text{joint}} = \log p(y, \tau, B_\tau) = \alpha \log p(y|\tau, B_\tau) + \log p(\tau, B_\tau)$, with $\alpha$ adjusting the weight of log-likelihood. RWS estimates ELBO by the difference between $\log p_{\text{joint}}$ and the log variational distribution for each sample:

$$\text{ELBO}_{\text{RWS}} = \log(\frac{1}{N} \sum_{i=1}^{N} \exp(\log p_{\text{joint}}) - \log q(\tau_i) - \log q(B_{\tau_i})), \tag{11}$$

Although straightforward, RWS may exhibit high variance when sample weights differ significantly. In contrast, VIMCO mitigates gradient estimation variance using Control Variates (CVs). For each sample $i$, $\bar{\text{cv}}_i$ is the average of CVs from all other samples:

$$\bar{\text{cv}}_i = \frac{1}{N-1} \sum_{j \neq i} \text{cv}_j, \quad \text{cv}_i = \log(\sum_{j \neq i} \exp(\text{s}_j + \bar{\text{cv}}_i) - \log N), \tag{12}$$

where $\text{s}_i = \log p_{\text{joint}} - \log q(\tau_i) - \log q(B_{\tau_i})$. To further reduce bias introduced by CVs, VIMCO corrects the ELBO estimate by normalizing exponentiated differences:

$$\text{ELBO}_{\text{VIMCO}} = \log(\frac{1}{N} \sum_{i=1}^{N} \exp(s_i)) + \sum_{i=1}^{N} (\log(\frac{1}{N} \sum_{i=1}^{N} \exp(s_i)) - \text{cv}_i) \cdot \log q(\tau_i). \tag{13}$$

To further enhance gradient stability, we incorporate the LAX model (Grathwohl et al., 2017), utilizing a differentiable surrogate function to approximate complex log-likelihood and prior terms, thus mitigating gradient discontinuities caused by discrete sampling. A 2-layer MLP generates latent representations $z_\chi$ from node features, providing inputs for the LAX model. We then derive the surrogate gradient estimator by combining joint log-probability with VIMCO lower bound:

$$\hat{g}_{\theta,\text{LAX}} = (\nabla_\theta \log Q_\theta(\tau, B_\tau)) \cdot (\log p_{\text{joint}} - s_\chi(z_\chi)) + \nabla_\theta s_\chi(z_\chi)), \tag{14}$$

where $s_\chi(z)$ is the surrogate function approximating the target function $\log p_{\text{joint}}$. The LAX model's loss function is defined as:

$$\mathcal{L}_{\text{LAX}} = -\mathbb{E}_{q(\mathbf{z}_\chi)}[\log q(\tau) \cdot (\log p_{\text{joint}} - s_\chi(z_\chi)) + s_\chi(z_\chi) - \log q(B_\tau)], \tag{15}$$

combines with VIMCO to form the VI loss: $\mathcal{L}_{\text{VI}} = -\text{ELBO}_{\text{VIMCO}} + \mathcal{L}_{\text{LAX}}$. The final loss function $\mathcal{L}$ for the DART task is defined as:

$$\mathcal{L} = \lambda \mathcal{L}_{\text{DON}} + \mathcal{L}_{\text{VI}}, \tag{16}$$

where $\lambda$ is a hyperparameter. To ensure the stability of parameter updates during gradient descent, we also incorporate gradient clipping.

## 4 EXPERIMENTS

In this section, we demonstrate the effectiveness of our proposed MDTree as follows:

| RQ1: Performance | How well does MDTree perform in generating tree topologies (TDE) and inferring branch lengths (VBPI)? |
| --- | --- |
| RQ2: Time Efficiency | How efficient is MDTree in reducing runtime? |
| RQ3: Tree Quality | How optimal is MDTree to generate a tree structure? (RQ3-1)
How diverse are the tree topologies generated by MDTree? (RQ3-2)
How consistent is the MDTree-generated tree compared to MrBayes? (RQ3-3) |
| RQ4: Module Impact | How does each MDTree's module affect its performance? (RQ4-2)
How do key hyper-parameters affect MDTree? (RQ4-2) |
| RQ5: Case Study | What evolutionary relationships between species does MDTree learn? |

### 4.1 EXPERIMENT SETUP

**Evaluation Tasks and Datasets.** We assess MDTree's performance on two key tasks: TDE, which focuses on optimizing tree topologies with MLL metric, and VBPI, where tree topologies and branch lengths are jointly inferred, using ELBO and MLL. These evaluations span eight diverse benchmark datasets, covering various organisms like marine animals, plants, bacteria, fungi, and eukaryotes, as outlined in Appendix C.

**Baselines.** MDTree is compared against three primary groups of baselines: (1) MCMC-based methods (MrBayes (Ronquist et al., 2012), SBN (Zhang & Matsen IV, 2018a)), (2) Structure Representation methods (VBPI (Zhang & Matsen IV, 2018b), VBPI-GNN (Zhang, 2023)), which leverage pre-generated topologies, and (3) Structure Generation methods without pre-selected topologies. Notably, ARTree (Xie & Zhang, 2024), a comparable autoregressive method like ours, is highlighted for comparison. All training details and hyperparameters are provided in Appendix E.

### 4.2 COMPARISON RESULTS ON BENCHMARKS (RQ1)

Table 1: Comparison of KL divergence ($\downarrow$) across eight benchmark datasets with different methods. **Boldface** for the highest result, **Text** for the second highest result of traditional methods.

| Methods | Dataset (#Taxa,#Sites) | DS1 (27,1949) | DS2 (29,2520) | DS3 (36,1812) | DS4 (41,1137) | DS5 (50,378) | DS6 (50,1133) | DS7 (59,1824) | DS8 (64,1008) |
|---|---|---|---|---|---|---|---|---|---|
| | Sampled Trees | 1228 | 7 | 43 | 828 | 33752 | 35407 | 1125 | 3067 |
| | GT Tress | 2784 | 42 | 351 | 11505 | 1516877 | 809765 | 11525 | 82162 |
| MCMC-based | SBN | 0.0707 | 0.0144 | **0.0554** | 0.0739 | 1.2472 | 0.3795 | 0.1531 | 0.3173 |
| | SRF | 0.0155 | **0.0122** | 0.3539 | 0.5322 | 11.5746 | 10.0159 | 1.2765 | 2.1653 |
| | CCD | 0.6027 | 0.0218 | 0.2074 | 0.1952 | 1.3272 | 0.4526 | 0.3292 | 0.4149 |
| | SBN-SA | 0.0687 | 0.0218 | 0.2074 | 0.1952 | 1.3272 | 0.4526 | 0.3292 | 0.4149 |
| | SBN-EM | 0.0136 | 0.0199 | 0.1243 | 0.0763 | 0.8599 | 0.3016 | 0.0483 | 0.1415 |
| | SBN-EM-$\alpha$ | **0.0130** | 0.0128 | 0.0882 | **0.0637** | **0.8218** | **0.2786** | **0.0399** | **0.1236** |
| Structure Generation | ARTree | 0.0045 | **0.0097** | 0.0548 | 0.0299 | 0.6266 | 0.2360 | 0.0191 | 0.0741 |
| | **Ours** | **0.0036** | 0.0129 | **0.0446** | **0.0216** | **0.5751** | **0.1591** | **0.0169** | **0.0634** |

Table 2: Evaluation of MLL ($\uparrow$) on eight benchmark datasets. VBPI and VBPI-GNN utilize pre-generated tree topologies during training, making **direct comparisons challenging**. **Boldface** highlights the highest result, **Text** denotes the second highest of structure generation methods, and **Text** indicates the second highest of MCMC-based methods. Numbers in parentheses represent standard deviation (std).

| Methods | Dataset (#Taxa,#Sites) | DS1 (27,1949) | DS2 (29,2520) | DS3 (36,1812 ) | DS4 (41,1137) | DS5 (50,378) | DS6 (50,1133) | DS7 (59,1824) | DS8 (64,1008) |
|---|---|---|---|---|---|---|---|---|---|
| MCMC-based | MrBayes | -7108.42 (0.18) | **-26367.57** (0.48) | -33735.44 (0.50) | -13330.44 (0.54) | **-8214.51** (0.28) | **-6724.07** (0.86) | -37332.76 (2.42) | **-8649.88** (1.75) |
| | SBN | **-7108.41** (0.15) | -26367.71 (0.08) | **-33735.09** (0.09) | **-13329.94** (0.20) | -8214.62 (0.40) | -6724.37 (0.43) | **-37331.97** (0.28) | -8650.64 (0.50) |
| Structure Representation | VBPI | -7108.42 (0.12) | -26367.72 (0.11) | -33735.10 (0.10) | -13329.94 (0.31) | -8214.61 (0.67) | -6724.34 (0.68) | -37332.03 (0.43) | -8650.63 (0.55) |
| | VBPI-GNN | -7108.41 (0.14) | -26367.73 (0.07) | -33735.12 (0.09) | -13329.94 (0.19) | -8214.64 (0.38) | -6724.37 (0.40) | -37332.04 (0.12) | -8650.65 (0.45) |
| Structure Generation | ARTree | **-7108.41** (0.19) | **-26367.71** (0.07) | **-33735.09** (0.09) | **-13329.94** (0.17) | **-8214.59** (0.34) | **-6724.37** (0.46) | **-37331.95** (0.27) | **-8650.61** (0.48) |
| | phi-CSMC | -7290.36 (7.23) | -30568.49 (31.34) | -33798.06 (6.62) | -13582.24 (35.08) | -8367.51 (8.87) | -7013.83 (16.99) | NA | -9209.18 (18.03) |
| | GeoPhy | -7111.55 (0.07) | -26379.48 (11.60) | -33757.79 (8.07) | -13342.71 (1.61) | -8240.87 (9.80) | -6735.14 (2.64) | -37377.86 (29.48) | -8663.51 (6.85) |
| | GeoPhy LOO(3) | -7116.09 (10.67) | -26368.54 (0.12) | -33735.85 (0.12) | -13337.42 (1.32) | -8233.89 (6.63) | -6735.9 (1.13) | -37358.96 (13.06) | -8660.48 (0.78) |
| | PhyloGFN | -7108.95 (0.06) | -26368.90 (0.28) | -33735.60 (0.35) | -13331.83 (0.19) | -8215.15 (0.20) | -6730.68 (0.54) | -37359.96 (1.14) | -8654.76 (0.19) |
| | **Ours** | **-7101.38** (0.07) | **-26357.96** (0.06) | **-33715.31** (0.10) | **-13322.10** (1.34) | **-8210.76** (0.23) | **-6713.13** (0.32) | **-37326.50** (1.39) | **-8645.07** (0.69) |

**TDE Task.** We compare the KL divergence to measure the difference between the model's generated tree topology distribution $q_\theta(\tau)$ and the true posterior $p(\tau)$: $\mathrm{KL}(p(\tau)||q_\theta(\tau)) = \sum_\tau p(\tau) \log \frac{p(\tau)}{q_\theta(\tau)}$. Tab. 1 shows that our MDTree consistently achieves lower KL divergence across all datasets compared to MCMC-based and structure generation methods. On complex datasets such as DS5 and DS6, it outperforms ARTree and SBN, demonstrating superior scalability. Even on smaller datasets like DS1 and DS3, the performance remains competitive, highlighting the model's robustness. The comparison with ARTree underscores the advantage of autoregressive models, including ours, particularly on larger, more complex datasets.

Table 3: Comparison of MLL and runtime (seconds) between Tree Generation Methods with RWS and VIMCO optimization technique optimized over 400,000 iterations.

| Methods | MLL | Runtime (s) |
|---|---|---|
| ARTree_rws | -7107.74 | 128.7 |
| **MDTree_rws** | -7103.71 | 75.0(↓41.72%) |
| ARTree_vimco | -7106.59 | 114.7 |
| **MDTree_vimco** | -7101.38 | 63.7(↓44.46%) |

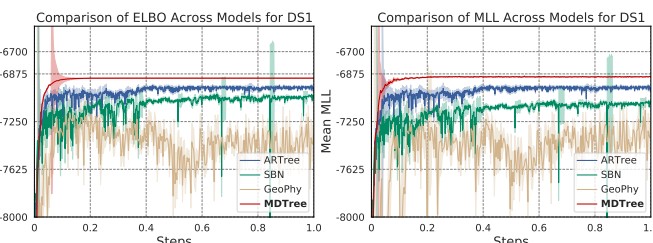

Figure 3: Comparison of ELBO. Figure 4: Comparison of MLL.

**VBPI Task.** We evaluate the VBPI task using ELBO and MLL metrics. Since direct computation of MLL is intractable, it is approximated via importance sampling. Unlike TDE, which relies on known tree topologies, VBPI evaluates the fit between model-generated tree topologies and branch lengths and the observed gene sequence data. As shown in Tab. 2 and Tab. 4, Tree Structure Generation methods exhibit broader applicability in MLL and ELBO metrics compared to Structure Representation methods, which are restricted by their reliance on pre-generated topologies. Our method, MDTree, consistently achieves the highest metrics across all datasets, highlighting its enhanced capacity to approximate the posterior distribution of tree topologies and branch lengths. Fig. 3 shows MDTree's superior stability and fast convergence in ELBO on DS1, outperforming baselines. ARTree and SBN improve later but with fluctuations, while GeoPhy performs the worst with consistently low and unstable values. Fig. 4 highlights MDTree's advantages in MLL, quickly reaching and maintaining high scores, whereas ARTree, SBN, and especially GeoPhy lag behind.

Table 4: Evaluation of ELBO (↑) on eight datasets. GeoPhy is not reported in the original publication that has been assessed by us.

| Methods | Dataset (#Taxa,#Sites) | DS1 (27,1949) | DS2 (29,2520) | DS3 (36,1812 ) | DS4 (41,1137) | DS5 (50,378) | DS6 (50,1133) | DS7 (59,1824) | DS8 (64,1008) |
|---|---|---|---|---|---|---|---|---|---|
| MCMC-based | SBN | -7110.24 (0.03) | -26368.88 (0.03) | -33736.22 (0.02) | -13331.83 (0.02) | -8217.80 (0.04) | -6728.65 (0.04) | -37334.85 (0.03) | -8655.05 (0.04) |
| Structure Generation | ARTree | **-7110.09** (0.04) | **-26368.78** (0.07) | **-33735.25** (0.08) | **-13330.27** (0.05) | **-8215.34** (0.04) | **-6725.33** (0.06) | **-37332.54** (0.13) | **-8651.73** (0.05) |
| | GeoPhy | -7116.67 (1.71) | -26434.84 (0.10) | -33766.72 (0.15) | -13389.36 (3.45) | -8220.91 (2.64) | -6769.41 (3.25) | -37882.96 (1.97) | -8654.39 (0.97) |
| | Ours | **-7005.98** (0.06) | **-26362.75** (0.12) | **-33430.94** (0.34) | **-13113.03** (3.65) | **-8053.23** (2.57) | **-6324.90** (1.26) | **-36838.42** (1.99) | **-8409.06** (1.09) |

## 4.3 Runtime Reduction and Efficiency Evaluation (RQ2)

MDTree demonstrates substantial runtime efficiency across all datasets, outperforming ARTree consistently. As shown in the right plot of Fig. 1, both runtime and the number of nodes are log-transformed on the vertical axes, with solid and dashed lines representing the RWS and VIMCO optimization techniques. MDTree achieves faster than ARTree across all datasets, with VIMCO providing further reductions, especially for MDTree-VIMCO, which exhibits the lowest runtime. The efficiency of MDTree becomes even more apparent as dataset complexity increases. Tab. 3 confirms this finding, with MDTree reducing runtime by 41.72% (RWS) and 44.46% (VIMCO) compared to ARTree while maintaining superior MLL metrics. This underscores MDTree's efficiency and scalability, particularly with VIMCO optimization.

## 4.4 Tree Parsimony in Phylogenetic Inference (RQ3-1)

To evaluate the parsimony of tree structures generated by the model, we follow established methodologies (Zhou et al., 2023a), minimizing the genetic mutations required to infer the optimal tree. The parsimony score evaluates how well the generated tree adheres to the principle of minimizing evolutionary changes, where fewer mutations are assumed to explain the observed genetic data better. We compare the results against the most parsimonious tree identified by the traditional PAUP* tool (Swofford, 1998). The parsimony score in Fig. 5 denotes the minimum mutations of genetic changes needed to account for the evolutionary relationships in the data. Since scores are plotted as negative values, lower scores indicate more complex trees and, consequently, poorer model performance. MDTree and ARTree achieved higher scores (approaching -4000) in fewer steps, reflecting simpler and more parsimonious trees. In contrast, PhyloGFN exhibited early fluctuations and ultimately stabilized around -5000, indicating suboptimal performance compared to others.

Table 5: Topological comparison of three tree diversity metrics. **Higher** values of Simpson's Diversity Index and the number of topologies accounting for the top 95% cumulative frequency indicate better diversity. In contrast, a **lower frequency** of the most frequent topology reflects a balanced distribution.

| Dataset | Statistics | MrBayes | ARTree | Ours |
|---------|-----------|---------|--------|------|
| DS1 | Diversity Index (↑) | 0.87 | 0.89 | **0.99** |
|     | Top Frequency (↓) | 0.27 | 0.1 | **0.007** |
|     | Top 95% Frequency (↑) | 42 | 10 | **121** |
| DS2 | Diversity Index (↑) | 0.89 | 0.96 | **0.99** |
|     | Top Frequency (↓) | 0.27 | 0.43 | **0.13** |
|     | Top 95% Frequency (↑) | 208 | 203 | **301** |
| DS3 | Diversity Index (↑) | 0.98 | 0.89 | **0.90** |
|     | Top Frequency (↓) | 0.02 | 0.01 | **0.004** |
|     | Top 95% Frequency (↑) | 753 | 509 | **1146** |
| DS4 | Diversity Index (↑) | 0.86 | 0.89 | **0.99** |
|     | Top Frequency (↓) | 0.11 | 0.05 | **0.002** |
|     | Top 95% Frequency (↑) | 4169 | 4125 | **8746** |

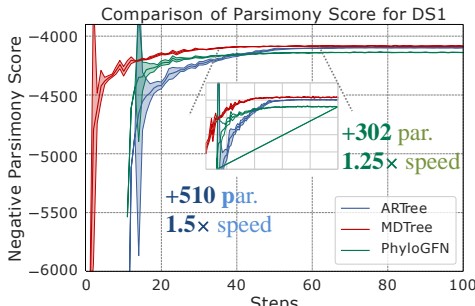

Figure 5: Comparison of negative parsimony scores on DS1 dataset. The parsimony score denotes the minimum number of variation steps required to interpret each tree. **The lower the negative score**, **the poorer the model performance**.

### 4.5 TREE TOPOLOGICAL DIVERSITY IN GENERATED TREES (RQ3-2)

To assess the diversity of tree topologies generated by MDTree, we use three metrics: Simpson's Diversity Index (He & Hu, 2005), Top Frequency, and Top 95% Frequency, as detailed in Tab. 5. A higher Diversity Index, which approaches 1, suggests broad diversity among generated tree topologies. A larger number of topologies in the Top 95% Frequency implies the generated trees are more varied and distributed across many unique structures. Conversely, a lower Top Frequency suggests the absence of a dominant tree structure, pointing toward a more balanced generation. For instance, in DS3, with 36 species sequences, the Top 95% Topologies metric reveals 1,146 distinct tree structures, indicating a wide range of possible phylogenetic solutions. MDTree achieves a Diversity Index close to 1, showcasing its capacity for generating highly diverse topologies even in complex datasets. Furthermore, the Top Frequency metric remains notably low, further reinforcing the diversity and indicating that no single tree topology is overly dominant.

### 4.6 BIPARTITION FREQUENCY FOR TREE QUALITY (RQ3-3)

In phylogenetic analysis, bipartition refers to dividing taxa (species or genes) into two groups on either side of a node within the tree. When multiple tree samples are generated, as in Bayesian inference methods like MrBayes, each sample may have a different topology. Bipartition frequency quantifies how often a specific bipartition appears across all tree samples, providing insight into the support for particular evolutionary relationships. We use this bipartition frequency distribution to assess the model's ability to capture phylogenetic relationships, as shown in Fig. 6. The horizontal axis indicates the bipartition rank within the tree topology, while the vertical axis displays the normalized occurrence frequency of each bipartition. The MDTree and MrBayes **curves are closely aligned**, indicating that MDTree's results closely match those of the widely accepted gold standard. In contrast, the ARTree method shows a noticeable deviation, especially in the higher-ranked bipartitions, demonstrating that MDTree offers

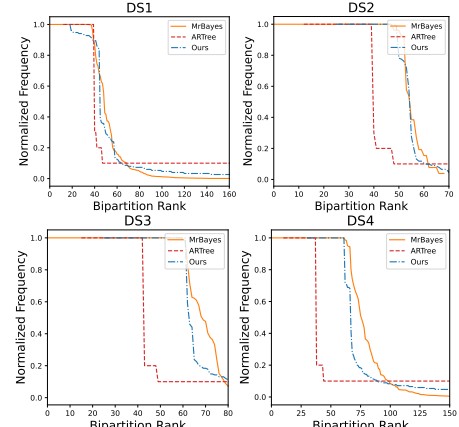

Figure 6: Bipartition frequency distribution of tree topologies. **The closer the two curves are, the better.**

improved accuracy over ARTree in capturing evolutionary structures. This suggests that MDTree captures the evolutionary patterns with greater accuracy compared to ARTree.

### 4.7 ANALYSIS AND ABLATION (RQ4-1)

Table 6: Study on different genomic LM.

| Method | MLL(↑) | ELBO(↑) |
|---|---|---|
| DNABERT2 | **-7101.38** | **-7005.98** |
| HyenaDNA | -7109.36 | -7014.17 |
| NT | -7111.07 | -7017.11 |

Table 7: Ablation study on four datasets.

| Method | DS1 | | DS2 | | DS3 | | DS4 | | Average |
|---|---|---|---|---|---|---|---|---|---|
| | MLL | ELBO | MLL | ELBO | MLL | ELBO | MLL | ELBO | |
| MDTree | -7101.38 | -7005.98 | -26357.96 | -26362.75 | -33715.31 | -33430.94 | -13322.10 | -13113.03 | **-20051.18** |
| w/o optimization | -7106.59 | -7010.34 | -26371.02 | -26374.01 | -33733.25 | -33447.94 | -13339.71 | -13130.01 | -20064.11 (-12.93) |
| w/ vimco w/o Lax | -7103.74 | -7007.86 | -26361.81 | -26368.52 | -33718.20 | -33436.07 | -13326.95 | -13118.60 | -20055.22 (-4.04) |
| w/o DON | -7105.05 | -7010.02 | -26366.47 | -26372.04 | -33723.67 | -33439.18 | -13332.38 | -13121.33 | -20058.77 (-7.59) |

We compare MDTree with three other schemes, yielding the following observations: (i) Removing optimization techniques like RWS or VIMCO led to a performance drop of 5.21 in MLL, as shown by slight fluctuations in the MLL curve in Fig. 7, highlighting their role in stabilizing convergence. (ii) Excluding the LAX model of VIMCO optimization caused a decrease of 2.36 in MLL and 1.88 in ELBO, indicating its effectiveness in reducing variance during discrete sampling. (iii) Tab. 6 and Tab. 7 show that the removal of the DON result in the most significant impact, with a drop of about 3.67 in MLL, underscoring its critical role in optimizing node addition order and improving tree generation. Overall, the full MDTree consistently achieves the best across both metrics. We select the genome-specific foundation model

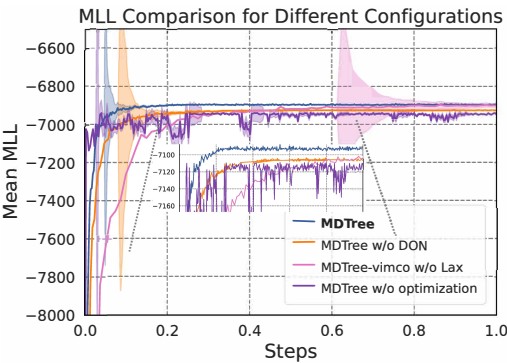

Figure 7: Ablation of different modules. MDTree w/o optimization curve exhibits **slight fluctuations**, emphasizing the importance of **optimization techniques** in improving stability.

DNABERT2 for our phylogenetic inference research. Although models like HyenaDNA (Nguyen et al., 2023) and Nucleotide Transformer (NT) (Dalla-Torre et al., 2023) excel in long-sequence modeling, they are less apt for our specific needs. As shown in Tab. 6, DNABERT2 outperforms others, likely due to its specific optimization for genomic data.

## 4.8 VISUALIZATION OF PHYLOTREE STRUCTURE ON REAL-WORLD DATA (RQ5)

To assess the biological relevance of the tree structure generated by MDTree, we applied it to construct a phylogenetic tree for an Angiosperms353 genomic dataset (Zuntini et al., 2024). The tree successfully recovered major branches within the order Rosales, revealing distinct evolutionary lineages, including Rosaceae, Moraceae, and Polygonaceae families. As shown in Fig. 8, the genera Polygala vulgaris and Polygala balduinii are clearly separated from other groups, consistent with their classification in the Potentillaceae family. The remaining groups, distinguished by color, represent genera within the Rosaceae and Moraceae families, such as Rosa, Rubus, Ficus, and Adansonia. In Rosaceae, genera like Rosa, Rubus, and Prunus highlight their common evolutionary ancestry, while in Moraceae, Ficus, and Broussonetia reflect the internal diversity and evolutionary divergence within the family.

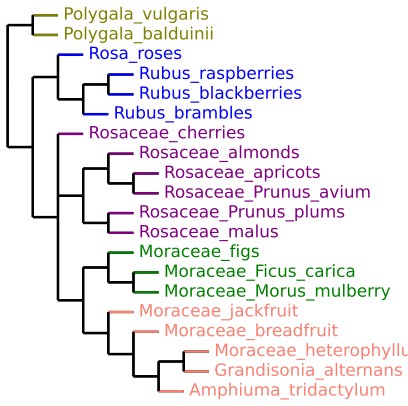

Figure 8: Visualization of Generated Trees on Angiosperms353.

## 5 CONCLUSION AND LIMITATION

**Contributions.** In this work, we introduce MDTree, a novel framework redefining phylogenetic tree generation as a Dynamic Autoregressive Tree Generation task. MDTree leverages a Dynamic Ordering Network to learn biologically informed node orders from genomic sequences, overcoming fixed or random order limitations. It integrates GNNs and Language Models to capture complex topologies, while a Dynamic Masking Mechanism enables parallel node processing, improving efficiency. Experiments show MDTree achieves SOTA performance on phylogenetic benchmarks. **Limitations and Future Work:** MDTree has yet to be applied to other sequence types, such as protein sequences. Future work will explore multimodal approaches and scaling the model for complex evolutionary scenarios. Additional details are in Appendix F.

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

SUPPLEMENT MATERIAL

# A  BACKGROUND

**Bayesian Methods**  Traditional statistical approaches like Maximum Likelihood Estimation (MLE) (Izquierdo-Carrasco et al., 2011; Solís-Lemus & Ané, 2016) and Bayesian Inference via Markov Chain Monte Carlo (MCMC) (Zhang et al., 2018; Wang et al., 2020) have been central to phylogenetic inference. However, as species numbers increase, these methods face severe computational limitations. The exponential growth in tree topologies—$(2N - 5)!!$ for unrooted bifurcating trees—leads to combinatorial complexity, making the simultaneous optimization of continuous branch lengths and discrete tree structures infeasible for large datasets. Additionally, MCMC-based methods, in particular, struggle with the multimodal nature of posterior distributions in high-dimensional tree spaces (Tjelmeland & Hegstad, 2001). Their reliance on local proposal mechanisms limits their ability to transition between distant peaks, leading to slow convergence and sampling inefficiencies (Whidden & Matsen IV, 2015; Zhang & Matsen IV, 2018b). Recent advancements, such as Variational Inference approaches, aim to improve efficiency but often involve simplifying assumptions that compromise the robustness of marginal likelihood estimation.

**Phylogenetic Posterior and Variational Inference (VI)**  Variational Autoencoders (VAE) Kingma & Welling (2013) is a deep generative model that learns input data distribution by encoding it into a latent space. In this process, the encoder maps each input $x$ to a latent space defined by parameters: mean $\mu$ and variance $\sigma$. Latent variables $z$ are then sampled from this distribution for data generation.

VI is employed within VAE to handle the computational challenges of estimating marginal likelihoods of observed data. This involves computing the log of the marginal likelihood:

$$\max_{\theta} \log p_\theta(X) = \sum_{i=1}^{N} \log \int_Z p_\theta(X, Z) \mathrm{d}z \tag{17}$$

where $p_\theta(X, Z)$ represents the joint distribution of the observable data $x$ e.g. a Gaussian distribution, $\mathcal{N}(x|\mu, \sigma)$ and its latent encoding $Z$ under the model parameter $\theta$.

Since the direct estimation of marginal likelihoods is typically infeasible, VI introduces a variational distribution $q_\phi(z|x)$ to approximate the true posterior. The goal of VI is to maximize the Evidence Lower Bound (ELBO), formulated as:

$$\mathrm{ELBO} = \mathbb{E}_{q_\phi(z|x)}[\log p_\theta(x|z)] - \mathrm{KL}[q_\phi(z|x)||p(z)] \tag{18}$$

The first term is the reconstruction log-likelihood, $\log p_\theta(x|z)$ can be considered as a decoder, i.e., the log-likelihood between the reconstructed data and the original data given the potential representation. The second term, the KL divergence, quantifies the difference between the variational posterior $q_\phi(z|x)$ and the latent prior $p(z)$. Usually, VAE utilizes a reparameterization trick for gradient backpropagation through non-differentiable sampling operations. Once trained, VAEs can generate new data by sampling directly from the latent space and processing it through the decoder.

# B  RELATED WORK

Phylogenetic inference methods can be broadly categorized into two major classes: traditional methods and deep learning-based methods. Each class can be further divided into graph structure generation and representation models. In this section, we review these approaches in detail.

## B.1  TRADITIONAL METHODS

Traditional phylogenetic inference methods primarily rely on predefined evolutionary models and statistical inference techniques. These methods typically assume specific evolutionary processes and use statistical approaches to search and optimize within a given tree structure space. They can be classified into graph structure generation and representation models.

**Graph Structure Generation Models:** MrBayes Ronquist et al. (2012) generates phylogenetic trees using Bayesian inference, estimating posterior probabilities based on sample relative frequency

(SRF). However, the high-dimensional combinatorial space poses accuracy challenges, particularly for low-probability trees, requiring large sample sizes for stability. VaiPhy Koptagel et al. (2022) introduces the SLANTIS sampling strategy Diaconis (2019) to generate tree structures by learning phylogenetic tree topologies. This approach combines basic biological models, such as the JC model, to estimate branch lengths, producing more accurate tree structures.

**Graph Structure Representation Models:** SBN (Structured Bayesian Networks) Zhang & Matsen IV (2018a) focuses on learning the probability distribution of tree topologies from existing phylogenetic trees. By modeling subsplit relationships within a given set of trees, SBN captures the probabilistic structure of the entire tree space without directly estimating branch lengths. VBPI (Variational Bayesian Phylogenetic Inference) Zhang & Matsen IV (2018b) builds on the tree topology probability distributions provided by SBN, using variational inference to estimate the posterior distribution of tree structures. This method further optimizes branch lengths, offering a precise approximation of the posterior distribution.

While traditional methods provide a solid theoretical foundation, they often struggle with the complexity of high-dimensional data and intricate evolutionary relationships. The emergence of deep learning has introduced new approaches to address these challenges.

### B.2 DEEP LEARNING-BASED METHODS

In recent years, deep learning techniques have demonstrated significant potential in phylogenetic inference, especially when dealing with complex, high-dimensional genomic data. These methods excel in generating and representing phylogenetic trees by learning latent representations or structural features from the data. They can be categorized into graph structure generation and representation models.

**Graph Structure Generation Models:**

- **Bayesian Generative Models (e.g., VAE):** These models learn latent representations of graphs using variational inference, from which new tree structures can be sampled. Geo-Phy Mimori & Hamada (2024) exemplifies this approach by leveraging VAE to model the latent space of phylogenetic trees, generating diverse structures that accommodate complex evolutionary histories.

- **Autoregressive Models:** Autoregressive models generate tree structures incrementally, making them suitable for tasks with well-defined sequences or hierarchies. ARTree Xie & Zhang (2024) employs a graph autoregressive model to generate detailed topologies, with branch lengths independently estimated using classical evolutionary models.

- **Diffusion Models:** Although diffusion models have not been widely applied in phylogenetic tree generation, our study integrates diffusion models with autoregressive models to generate the node addition order, enhancing the accuracy of tree structures. This demonstrates the potential of diffusion models in high-quality phylogenetic inference.

- **Generative Flow Networks (GFlowNets):** As illustrated by PhyloGFN Zhou et al. (2023a), GFlowNets Hu et al. (2023) combined with Markov decision processes optimize the generation path, progressively constructing complex phylogenetic tree structures.

**Graph Structure Representation Models:** VBPI-GNN Zhang (2023) leverages pre-generated candidate tree structures and SBN-provided tree topology probability distributions, combined with variational inference, to optimize branch lengths and tree topologies, ultimately providing a precise approximation of the posterior distribution.

## C DATASETS

Our model, MDTree, conducts phylogenetic inference on biological sequence datasets comprising 27 to 64 species, as compiled in Lakner et al. (2008). Importantly, our approach does not require sequences to be of uniform length, thereby addressing a common limitation in traditional phylogenetic analyses. Tab. A1 summarizes the statistics of the benchmark datasets.

Table A1: Statistics of the benchmark datasets from DS1 to DS8..

| Dataset | # Species | # Sites | Reference |
|---------|-----------|---------|-----------|
| DS1 | 27 | 1949 | Hedges et al. (1990) |
| DS2 | 29 | 2520 | Garey et al. (1996) |
| DS3 | 36 | 1812 | Yang & Yoder (2003) |
| DS4 | 41 | 1137 | Henk et al. (2003) |
| DS5 | 50 | 378 | Lakner et al. (2008) |
| DS6 | 50 | 1133 | Zhang & Blackwell (2001) |
| DS7 | 59 | 1824 | Yoder & Yang (2004) |
| DS8 | 64 | 1008 | Rossman et al. (2001) |

## D  METHOD

**Calculation of the number of unlabelled nodes in DON.** The number of nodes unmasked at each step is dynamically determined by a mask rate modulated by a cosine function. Given a total of $T$ steps and $U$ nodes to be unmasked per step, the proportion of nodes to be unmasked at each step $t$ is computed as follows: $r_t = \frac{t}{T}, t = 1, 2, \ldots, T$. This is modulated by a cosine function to produce the mask rate: $\text{mask\_rate}_t = \cos\left(\frac{\pi}{2} \cdot r_t\right)$, where $\text{mask\_rate}_t$ controls the relative number of nodes unmasked at step $t$. The final number of nodes unmasked at each step is normalized to ensure that the total number of unmasked nodes across all steps sums to $T \times U$: $\text{unmasked\_nodes}_t = \left\lfloor \frac{\text{mask\_rate}_t}{\sum_{t=1}^{T} \text{mask\_rate}_t} \cdot T \cdot U \right\rfloor$, where $\lfloor \cdot \rfloor$ denotes rounding to the nearest integer.

## E  EXPERIMENT

### E.1  TRAINING DETAILS

We focus on the most challenging aspect of the phylogenetic tree inference task: the joint learning of tree topologies and branch lengths. For this, we employ a uniform prior for the tree topology and an independent and identically distributed (i.i.d.) exponential prior (Exp(10)) for the branch lengths. We evaluate all methods across eight real datasets (DS1-8) frequently used to benchmark phylogenetic tree inference methods. These datasets include sequences from 27 to 64 eukaryote species, each comprising 378 to 2520 sites. For our Monte Carlo simulations, we select $K = 2$ samples and apply an annealed unnormalized posterior during each $i$-th iteration, where $\lambda_n = \min\{1.0, 0.001 + i/H\}$ acts as the inverse temperature. This parameter starts at 0.001 and gradually increases to 1 over $H$ iterations, effectively simulating a cooling schedule commonly used in annealing algorithms, similar to the approach in Zhang & Matsen IV (2018a), with an initial temperature of 0.001, which gradually decreases over 100,000 steps.

During the model training process, we utilize stochastic gradient descent to process a total of one million Monte Carlo samples, employing $K$ samples at each training step. The stepping-stone (SS) algorithm Xie et al. (2011) in MrBayes is viewed as the gold-standard value. All models were implemented in Pytorch Paszke et al. (2019) with the Adam optimizer Kingma & Ba (2014). The MLL estimate is derived by sampling the importance of 1000 samples, with the larger mean value being better. The learning rate is initially set to 1e-4 and is reduced by 0.75 every 200,000 training steps. Momentum is set at 0.9 to prevent the optimization process from becoming trapped in local minima. Utilizing the StepLR scheduler, the current learning rate is multiplied by 0.75 every 200,000 steps to ensure steady progression, detailed in Tab. A2.

Table A4: Hyperparameter Analysis of MDTree Performance.

| Configurations | Parameters | | | |
|---|---|---|---|---|
| DON_hd=32, Tree_hd=100 | | | | |
| # Heads | 1 | 2 | 3 | 4 |
| ELBO | -7517.98 | -7111.95 | -7106.65 | **-7005.98** |
| MLL | -7333.14 | -7116.65 | -7104.82 | **-7101.38** |
| DON_hd=32, Tree_hd=100 | | | | |
| \alpha | 0.025 | 0.05 | 0.1 | 0.15 |
| ELBO | -7108.67 | **-7005.98** | -7114.14 | -7112.40 |
| MLL | -7107.32 | **-7101.38** | -7110.38 | -7115.27 |
| # Heads=4, Tree_hd=100 | | | | |
| DON_hidden dim | 8 | 16 | 32 | 64 |
| ELBO | -7016.75 | -7011.16 | **-7005.98** | -7013.96 |
| MLL | -7113.84 | -7117.83 | **-7101.38** | -7105.18 |
| # Heads=4, DON_hd=32 | | | | |
| Tree_hidden dim | 500 | 100 | 150 | 200 |
| ELBO | -7013.71 | **-7005.98** | -7012.07 | -7008.93 |
| MLL | -7112.71 | **-7101.38** | -7102.05 | -7121.51 |

Table A2: Training Settings of MDTree.

| Training Configuration | |
|---|---|
| Optimizer | Adam optimizer |
| Learning rate | 1e-4 |
| Schedule | Step Learning Rate |
| Weight Decay | 0.0 |
| momentum | 0.9 |
| base_lr | 1e-4 |
| max_lr | 0.001 |
| scheduler.gamma | 0.75 |
| annealing init | 0.001 |
| annealing steps | 400,000 |

Table A3: Hyperparameters for MDTree.

| DON | |
|---|---|
| Hidden Dim. | 32 |
| # Layer | 2 |
| Output Dim. | 1 |
| $\alpha$ of Equ. 5 | 0.05 |
| TreeEncoder | |
| Hidden Dim. | 100 |
| # Heads | 4 |
| DGCNN | |
| # Layer | 2 |

### E.2 HYPER-PARAMETER ANALYSIS (RQ4-2)

Tab. A4 summarizes the hyperparameter search results for DON hidden dimension, Tree Network (Transformer) hidden dimension, and the number of attention heads. When increasing the number of heads from 1 to 4, ELBO improves from -7517.98 to -7005.98, and MLL improves from -7333.14 to -7101.38, demonstrating that more attention heads allow the model to capture richer dependencies. For the DON hidden dimension, a value of 32 achieves the best results, with an ELBO of -7005.98 and MLL of -7101.38. Similarly, tuning the Tree hidden dimension shows that 100 is optimal, yielding an ELBO of -7005.98 and MLL of -7101.38, while further increasing the dimension does not result in better performance. These results highlight the importance of tuning the number of heads and hidden dimensions to balance model complexity and generalization.

### E.3 VISUALIZATION OF PHYLOTREE STRUCTURE ON REAL-WORLD DATA (RQ5)

To evaluate the biological relevance and performance of MDTree, we compared the phylogenetic trees generated by MDTree and ARTree on the Angiosperms353 dataset (Zuntini et al., 2024). As shown in Fig. A1, the tree generated by MDTree accurately clusters species from the same genera and families. For instance, in the Rosaceae family, genera like Rosa, Rubus, and Prunus are grouped together, reflecting their common evolutionary ancestry. Similarly, in the Moraceae family, Ficus and Morus are placed close to each other, highlighting their evolutionary divergence within the same lineage. The distinct classification of Polygala vulgaris and Polygala balduinii further validates the biological significance of the tree, consistent with their classification in the Potentillaceae family. In

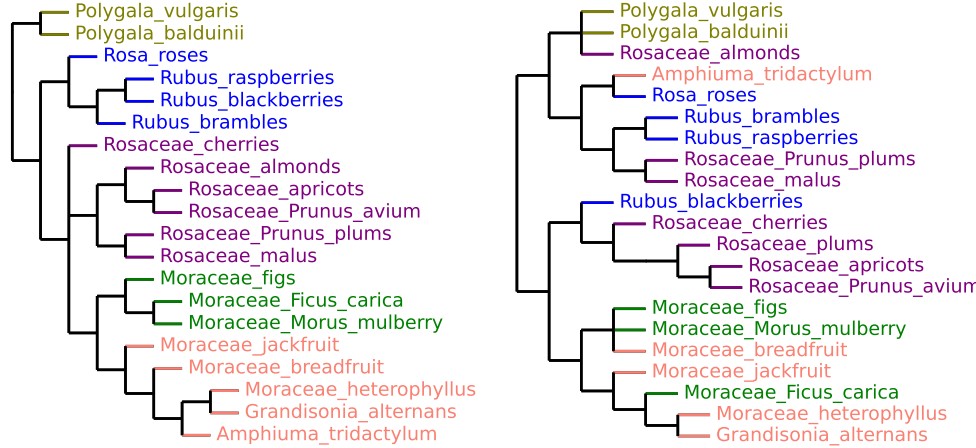

Figure A1: MDTree Visualization of Generated Trees on Angiosperms353 dataset.

Figure A2: ARTree Visualization of Generated Trees on Angiosperms353 dataset.

contrast, as shown in Fig. A2, the ARTree-generated tree demonstrates less biological coherence. For instance, certain genera within Rosaceae and Moraceae are incorrectly clustered, disrupting the phylogenetic structure and lineage relationships.

## F    LIMITATION

While MDTree demonstrates significant advances in phylogenetic inference, including improved accuracy and efficiency in tree structure generation, several limitations remain. First, MDTree leverages pretrained genomic language models to initialize sequence representations, which enhances performance. However, alternative representations, such as one-hot encoding, can also be used, albeit with reduced accuracy and efficiency. Additionally, MDTree has so far been validated only on genomic sequences and has not been extended to other sequence types, such as protein sequences or non-biological data. Finally, while MDTree improves computational efficiency, its performance on extremely large-scale datasets or sequences with complex multimodal dependencies, such as integrating genomic and proteomic data, remains unexplored.

