# OpenReview forum: "MDTREE: A Masked Dynamic Autoregressive Model for Phylogenetic Inference"
_ICLR.cc/2025/Conference — Submitted to ICLR 2025_

### Official Review · Reviewer_oF9c · 2024-10-27

**Soundness:** 2
**Presentation:** 3
**Contribution:** 3
**Rating:** 6
**Confidence:** 5

**Summary:**

The manuscript introduces MDTree, a novel approach to phylogenetic tree inference. MDTree addresses the issues of model complexity and computational efficiency. By leveraging a Diffusion Ordering Network, MDTree dynamically learns the optimal node order, enabling more accurate and efficient tree construction. This approach incorporates graph neural networks (GNNs) and language modeling techniques to capture complex evolutionary relationships. Additionally, a dynamic masking mechanism allows for parallel node processing, further accelerating the inference process. The authors benchmark the performance in several aspects to show the effectiveness of MDTree.

**Strengths:**

The authors proposed a novel method and conducted a comprehensive evaluation by comparing MDTree with several baseline methods across various datasets and metrics.

**Weaknesses:**

The paper's experimental evaluation is hindered by several aspects. Firstly, the parameter settings for baseline methods are not well-documented, potentially impacting the strength of reported performance. Secondly, the absence of publicly available code limits reproducibility and hinders independent verification of the results. Additionally, the methods compared in each table are inconsistent, lacking clear explanations for these choices. For example, while MrBayes is included in Table 2, it is absent from Table 1, raising questions about the rationale behind these decisions.

While the paper introduces a novel approach to phylogenetic tree inference, the literature review in the introduction appears to conflate different concepts. For instance, the discussion of Solis-Lemus & Ané, 2016 and Zhang et al., 2018, which focus on network inference from gene trees/multiple sequence alignments under the multispecies network coalescent model, seems to be mixed with the concept of gene tree inference from sequence data, the primary focus of the proposed MDTree method. A clearer distinction between these approaches would enhance the paper's clarity and contextual understanding.

Besides the major concerns, below are some minor concerns.

Figure 1: left and right are opposite. Run time unit is missing.

The name of the proposed method is not consistent in tables. E.g., Table 1: MDTree, Table 2: Ours.

**Questions:**

My suggestion is to address the weaknesses above.
1. describe baseline method settings
2. provide code availability to reproduce the result
3. compare all methods in each metric or explain why a certain method is not included
4. review the related work and discuss existing method and gap in a more clear way
5. proofread the paper

---

### Official Review · Reviewer_ESW1 · 2024-11-03

**Soundness:** 3
**Presentation:** 3
**Contribution:** 4
**Rating:** 8
**Confidence:** 5

**Summary:**

This paper introduces a new framework for phylogenetic inference MDTree. Traditional methods like MCMC and previous deep learning methods are limited by their high computational cost, low efficiency and low inference accuracy. The new framework uses multiple techniques to effectively resolve these limitations, including a diffusion ordering network (DON) to generate node orders according to evolutionary relationships, autoregressive constriction module with dynamic masking mechanism to generate tree structures in parallel. The model uses a dual-pass traversal to estimate the tree branch lengths.
This study includes an extensive evaluation which indicate that MDTree has a robust performance on datasets with variant taxa number and sequence length. Its computational cost and running time outperformed the state-of-the-art methods. This study also includes a comprehensive ablations study on models and hyper parameters to demonstrate the contribution and robustness of the modules.

**Strengths:**

This is a quite impressive work that solved multiple pain points in phylogenetic inference.
The idea proposed by this paper is innovated and effective, improves the phylogenetic tree inference on both efficiency and accuracy.
The paper is well organized with a clear writing style. It is easy to follow the author’s idea.
The experiment design is comprehensive. The author considered about multiple aspects of phylogenetic inference, such as running time, tree quality, model robustness, empirical study. The results are convincing.

**Weaknesses:**

There are certain weakness about this study. The complex architecture and multi-layered optimization requirements may limit the practical application. It is worth to consider about pack the framework into a user friendly package or online service. This will not only help people who are interested in this study, but also increase the impact of this impressive work.
There are some details about the method and the evaluation metrics are ignored in the paper, such as how does DON determine the node order based on genomic embeddings? What is the impact of sequence divergence and species evolutionary relationship distance to the node order and inferred phylogenies? Why generating highly diverse tree topology is necessary, especially in biological analysis.

**Questions:**

1. How does DON determine the node order based on genomic embeddings? How much does it impact the final inference if the species sequence order differs?
2. How does the mask rate selection impact the parallel computation of nodes insertion and overall model running efficiency?
3. There is no summarization of of the sequence divergence and evolutionary relationship distances about the dataset used in this study. It is necessary to evaluate the impact of sequence divergence on the model performance. The author can also consider adding simulated datasets experiments to better control the sequence divergence.
4. What is the purpose of generating highly diverse tree topologies in biological research? What type of  practical application needs such diverse tree topology instead of a highly confident and accurate phylogenetic tree?
5. Consider adding bootstrap analysis for phylogenetic support estimation to better indicate how confident is the inferred phylogenies.

---

### Official Review · Reviewer_i3bg · 2024-11-03

**Soundness:** 3
**Presentation:** 3
**Contribution:** 3
**Rating:** 6
**Confidence:** 2

**Summary:**

The authors present MDTree, a technique for inferring phylogenetic trees (topology + branch lengths) from a set of genomic sequences. To motivate their model, the authors reframe phylogenetic tree construction from the perspective of DART: dynamic autoregressive tree generation, which differs from its autoregressive predecessors by incorporating a node order learning step. To this end, MDTree uses a Diffusion Ordering Network (DON) using genomic language model embeddings to sort sequences. This enables better autoregressive generation and even makes it possible to add nodes in parallel. The authors benchmark MDTree on 8 classic phylogenetics datasets, comparing it to classical MCMC, structure generation, and autoregressive methods. In almost all benchmarks, they show state-of-the-art performance as well as improvements in secondary properties like computation speed, parsimony, and diversity.

Edit: updated score from 5 to 6 following discussion with the authors.

**Strengths:**

* **Strong benchmark performance** is the main strength of this paper. Across almost all dataset + task benchmarks, the authors claim state-of-the-art performance. This is shown in Tables 1--4 and Figures 3--4.
* **Extensive secondary benchmarks** characterize MDTree's runtime reduction, parsimony, topological diversity, and bipartition frequencies compared to ARTree (and sometimes other models).
* **Biological ordering** is a desirable addition to autoregressive models, ensuring that the phylogenetic tree construction task can aggregate as much information across nodes of the tree as possible. This addresses a key limitation of previous autoregressive methods.
* **Parallelism**, enabled by the biological ordering, is a desirable property for a computational method and appears to improve processing speeds substantially (as shown in Figure 1).
* **Use of embeddings** eliminates the restriction that all sequences are the same length common in other models; it is also likely to unlock improvements in MDTree "for free" as better genomic foundation models are trained.

**Weaknesses:**

* **Relationship to ARTree** is somewhat unclear, and although comparisons favor MDTree, the ARTree score is oftentimes quite close.
  * The authors should make it explicit what distinguishes MDTree from ARTree.
  * The ablations should make it clear which ablated forms of MDTree (if any) are worse than base ARTree
  * Since the benchmark scores of MDTree and ARTree are often quite close together, I am less impressed by "state of the art" results. If the authors could convince me why this position is mistaken, and their method is a *significant* improvement over ARTree, I would be amenable to improving my score.
* **Lack of motivation** for specific architectural choices. Most notably, the diffusion ordering network (DON) is justified in terms of the limitations of other autoregressive methods like ARTree; however, the specific choice of architecture is presented as arbitrary/self-evident. To this end, I have several questions:
  * What other options have the authors considered/tested? Why was the DON ultimately chosen?
  * How does the DON compare to a simple baseline that produces biologically meaningful orders without relying on deep learning? The authors may have a better sense of what a good baseline might be. However, I propose the following baseline as a reasonable starting point:
    1. Compute pairwise edit distances between genomic sequences (e.g. Levenshtein distances)
    2. Perform agglomerative clustering on the pairwise distances to get a crude tree
    3. Use an inorder traversal of the resulting tree to sort the leaves. This is your input order.
  * You cite "evidence of robustness across different taxa orders" in ARTree (line 163), but here you simply say "the influence of node orders on phylogenetic accuracy has not been thoroughly examined." The ablation-based evidence presented in Table 7 suggests that node order has a weak influence on model performance, but it would be more convincing to see a non ablation-based characterization (e.g. what is the variance in MLLs for random permutations of node order?)
* **Unintuitive choice of representations** to seed the DON. It is not apparent that genomic LM representations are the best candidates here, as the LMs are not actually trained to estimate evolutionary distances. Moreover, vector space representations of genetic sequences will always incur distortion, as the geometry of phylogenetic trees is inherently non-Euclidean as a result of the four-point condition.
* **The DART formulation** seems unnecessary. What is the advantage of reformulating phylogenetic tree construction (which we have a perfectly good description of already: learning a topology and a set of branch lengths), besides that it attempts to justify the use of a DON? If that is all, I would argue that "proper node orders improve phylogenetic inference" is a sufficient claim.
  * If other problems in phylogenetics are better viewed from the DART perspective, I would be interested in such an example. This would go a long way towards changing my mind on the value of this part of the paper.
* **Presentation** is unrefined throughout:
  * Figures are often cramped, and combined into figures with no clear logic (e.g. Figure 1 includes a cartoon and a runtime comparison)
  * Model details are crammed in pages 4 and 5. It is unclear without substantial cross-referencing and careful reading how all of the pieces fit together. While I understand the need to fit within the page limit, I would be interested in seeing the full architecture described in the Appendix.
    * "Mask rate modulated by a cosine function" (200) seems to be an essential detail of the autoregressive tree, but the equation is not given anywhere
* **Related work** does not discuss Bayesian methods since VBPI (except for VaiPhy). There have been many developments in this field since then.
* **Missing experiments**: oftentimes, certain models are missing from evaluations. For instance, MrBayes and structure representation models are missing from Table 1; many models are missing from Table 4; comparisons in terms of runtime, diversity, bipartition, etc., are only run for 2-3 models at a time. It is possible that these results are infeasible to generate, but the authors should make this explicit.

**Questions:**

* Why is it better that "closely related species should be placed earlier in the tree" (175) versus, e.g. simply clustering together? Is this robust to all topologies? For instance, what happens if you have two very distantly related subtrees, each of which has many species who are closely related to another?
  * Similarly, I am interested in worst-case performance of the clustering algorithm. For instance, if you had a linear tree, would you still be able to parallelize your algorithm?
* What should the reader take away from the Angiosperm tree in Figure 8? How does this compare to/improve on the trees generated by other models?
* Bayesian phylogenetics methods will typically include a proof that their estimators are consistent and unbiased. Is it possible to do something similar in the case of this method? If not, the authors should justify why it is worth abandoning such guarantees in favor of their model
* Will the model be made publicly available? If so, is it easy to use and install? Does it work on a variety of machines and operating systems?

---

### Official Review · Reviewer_5UDw · 2024-11-03

**Soundness:** 2
**Presentation:** 1
**Contribution:** 3
**Rating:** 5
**Confidence:** 4

**Summary:**

The paper proposes a new node ordering network to be able to better utilise autoregressive models to generate phylogenetic trees.

**Strengths:**

- Several new ideas on the technical side.
- (Marginally? hard to judge with the units and lack of error bars) better results.
- Improvements in run-time.

**Weaknesses:**

TLDR: I am willing to raise my score if the results are clarified and presented more convincingly (1. error bars 2. a clearer results table and figures 3. a better presentation of the baselines); the method is presented in a way that is much much clearer (I think starting from the problem then objective/loss function then the 3 main components, etc might help); and the method is better placed in the related work with a better discussion of the limitations. From my limited understanding of the paper it seems the technical contribution is there (albeit a little hard for me to judge), but the presentation is still far from the quality necessary for acceptance.

It should be said that I cannot judge whether the baselines are chosen appropriately.

Weaknesses:
- There are no statistical error margins (e.g. standard deviation) for the results. This is okay if the computational cost is huge (e.g. often the case in LLMs) if so please state this clearly as well as the running cost in GPU hours, the GPUs used, etc.
- Table 2 confuses me. For instance for DS1: three are numbers are bold, but not even the 3 highest ones (when MLL higher is better according to the caption), e.g. VBPI-GNN also has the third highest score -7108.41. Also the difference between some of the results seem incredibly marginal. Furthermore, negative MLL might be clearer rather than having a minus sign in front of every number.
- The method has many components, which on the hand is impressive that the authors managed to build this system and make it work, but also comes with limitations that aren't adequately discussed in my opinion. There are a great number of hyperparameters, but far too little space is dedicated to ablating them or acknowledging the difficulty of choosing them.
- The related work is quite brief given that the model borrows many techniques related work it compares against. A clearer delineation would be helpful to the reader.

Clarity:
- The way indices i and t are re-used is confusing.
- It is imperative to give the citation of each baseline method considered. In the paper they are merely named, but not cited. This allows for confusion if two methods share the same name for instance and is generally poor practice.
- The Table captions could be improved, what are the numbers in paranthesis in Table 2? What is the grey background mean? Why are multiple numbers in bold per dataset?
- The y-axis range in Figure 3 makes the results really hard to discern.

Minor:
- Line 164 "As discussed, ..." please add a link to where this is discussed, this helps non-linear reading, which is standard for papers.
- Equation 1 LHS says h_i, the text says h_t

**Questions:**

- Why is G_t a graph? I would have thought it is a DAG.
- How do you define \tau and B_\tau mathematically?
- The mapping F takes a single species sequence to a tree topology, but in the text it states that F depends on G, which is not reflected in the notation. In addition, why is each species sequence sent to a separate tree topology? There is only a single evolutionary timeline we want to analyse.
- Line 186 beta increases montonically as what value goes from 0 to 1?
- What positional encoding function PE is being used?
- Is the categorical distribution in Eq 3 the forward process of the diffusion process?
- What is the MLL metric? Please either expand the acronym or give a citation.
-Table 3: What was the hardware used?
- I presume alpha in Equation 5 is highly sensitive to N? How come the authors choose to take the softmax of a softmax rather than directly adding the alpha term to L_i?
- What is the importance of branch lengths?
- You have 3 distinct components, could you clarify how the gradient flows? I presume that at the boundary between the modules the gradient is stopped due to the discrete decision boundary? If so, how does the ordering network for instance get any signal.
- What modules are pretrained? Which are trained from scratch? How do you initialise the weights?

---

### Official Review · Reviewer_DTGp · 2024-11-04

**Soundness:** 3
**Presentation:** 1
**Contribution:** 3
**Rating:** 5
**Confidence:** 2

**Summary:**

This paper introduces a new method for phylogenetic tree inference which extends beyond autoregressive models and is claimed to improve both computational efficiency and accuracy. Specifically, it focuses on learning a better ordering strategy for adding a new node into the phylogenetic tree from GLM priors as opposed to using fixed orders (lexicographical order) in autoregressive models. The authors framed the problem as masked dynamic autoregressive tree generation and introduced a Dynamic Ordering Network (DON), which is an absorbing diffusion model for learning the node adding order from pre-trained genomic language models (GLMs) to better leverage the biological and evolutional information. They further introduced several techniques for efficiency improvement, including dynamic masking and parallel processing, dual-pass tree traversal for branch length estimation, and LAX model for variance reduction. Extensive experiments show improved accuracy and efficiency of the proposed framework.

**Strengths:**

The paper studies an important problem in bioinformatics concerning phylogenetic tree inference. The main highlight is the introduction of DON which infers node orders and enables sampling multiple nodes for parallel computation, which makes use of the strong prior in pretrained GLM and might be more flexible than the fixed order AR method.

The method is novel and adds a significant improvement to AR method. Additional techniques are introduced to further improve efficiency, generation consistency, and optimization. The perspective of studying the influence of node orders on phylogenetic accuracy is also novel.

The authors conduct extensive experiments across multiple tasks and datasets related to phylogenetic tree inference and consistently outperform the baselines including ARTree which is likely the previous SOTA. They also showed strong metrics on computational efficiency, and a thorough ablation study showing the importance of DON.

**Weaknesses:**

The paper studies a very specific task, phylogenetic tree generation problem, within the bioinformatics domain. Although the task might be important in the domain, the introduced methodology seems to be highly specified for this problem alone, which might limit its significance in the general graph generation area.

The biggest concern lies in the writing clarity, particularly the DON description. Multiple key information is missing and several notations are inconsistent across the main text. Firstly, the DON module seems to assume some graph structure already known among the sequence (e.g., figure 2.A has a ring structure). How is this graph constructed? It cannot be the tree structure as the phylogenetic tree has not been generated yet at this stage.

The presentation of DON in 3.1 can be largely improved, with many key notations and parameters unexplained. The biggest gap is the lack of a proper definition of the forward and backward diffusion process, with clear correspondence to time t. It starts with directly “updating node features $h_t$” without defining what t means. It is also not clear what positional encoding $PE_t(g_i)$ means, it is used with subscription $t$ but isn’t the position of node i fixed? How does PE vary with time t and why is it varying with t? It is not clear whether the transition probability in (2) defines a forward corruption process from t=N to 0 or t=0 to N? How can we make sure only a single node is selected to be absorbed at each time step? The notation of $h_t$ is also confusing, is it a single node embedding or embedding matrix for all nodes? There is a mixed use of $h_t$ and $h_i$.

The node order generation process after the entire graph is absorbed is also not explained well. Equation (3) defines a conditional probability between node embeddings $q(h_t|h_0, h_{(<t))}$, how can this be used for order determination? Shouldn’t one predict the probability of unmasking a node in a diffusion setting? It seems the transition matrix Qt only allows jumping from a non-masking state to a mask. How can this be reused for computing a cumulative transition matrix in the opposite direction (i.e. from masked to unmasked)?

Finally, it is not clear whether the DON is trained (e.g., with a certain score matching loss, and if so what is the training target given that optimal order is not available ahead of time?), or it is just a hand-crafted discrete forward diffusion process which is completely determined by the hyperparameters $\beta_{t,i}$. There is no description regarding how network parameters of the relational graph convolutional network used for node feature computation are trained either. There is a large discrepancy between what is described in 3.1 and training loss (10) in 3.4, where $q_\sigma(\sigma_t|G_0,\sigma_{(<t)})$ suddenly appears without definition.

In section 3.2 tree construction, a multi-head attention block with a query matrix Q is introduced MHA($Q, h_i, h_i$), what is the goal of Q here? It is initialized to an Identity matrix with size (N-3)*100, but was not mentioned later.

There are several typos and inconsistencies in naming terminology. E.g., the DON is sometimes referred as Diffusion Ordering Network and sometimes Dynamic Ordering Network.

**Questions:**

1.	How does node count measure runtime efficiency in figure 1 and why a lower node count is preferred?
2.	How do we get the initial graph structure that is used as input to DON?
3.	Is DON a completely separate and preceding step from the dynamic AR tree construction? Or the order determination step is roll out iterative after each node insertion step?
4.	See other questions in Weakness

---

### Official Review · Reviewer_zB4G · 2024-11-05

**Soundness:** 3
**Presentation:** 4
**Contribution:** 3
**Rating:** 6
**Confidence:** 3

**Summary:**

The paper provides a new deep learning based method that incorporates language model to extract biological priors to find a node insertion ordering. They improve on state of the art methods using autoregressive models and provide comprehensive experiments.

**Strengths:**

- The paper improves on existing methods, notably ARTree, for deep learning for phylogenetic inference.
- The proposed central problem, that of finding a proper taxon insertion order is an important piece of any phylogenetic inference algorithm and deserves more highlights in deep learning-based approaches.
- The use of language models to extra biological priors is quite novel and general.
- Experiments are relatively extensive, at the base line of deep learning-based approaches.

**Weaknesses:**

- The paper's main contribution is methodological but compared to the most closely related method, ARTree, there are only marginal improvements across datasets. This is also in light of the fact that the proposed methodology is extremely more computational intensive, both in terms of runtime and carbon footprint. With so much more computation, it is not too unfair to expect a more pronounced difference. Perhaps it is advisable to find conditions where dynamic node ordering strongly affects the tree reconstruction methods. If it is too hard to find such conditions, perhaps it is not as bad a problem as stated in the paper.
- The paper's key insights (compared to literature) is a method to learn an insertion ordering of the taxa. However, it is not clear that the proposed methodology to find such an ordering is clearly advantageous compared to other different orderings. The baseline considered against use a lexicographical ordering, which is just arbitrary. What happens when a different ordering is used?
- Related to the topic of choosing the right taxa ordering: theoretically, given just one correct planar ordering of the taxa (draw the true tree onto the plane and number leaves from left to right, there is a trivial greedy algorithm to find the correct tree structure and branch length from tree distance approximated from DNA sequences. As a result, find the correct order is one of the hardest subproblem of tree inference.
- There are other line of work that uses Prim ordering of the distance matrix between taxa as the ordering to add taxa into the tree and implemented with maximum likelihood heuristics (Zhang, Rao, Warnow 2019 Constrained incremental tree building: new absolute fast converging phylogeny estimation methods with improved scalability and accuracy; Le et al. 2021 Using Constrained-INC for Large-Scale Gene Tree and Species Tree Estimation). These are not deep learning-based methods so it's not directly comparable, but at least a discussion on the existing orders that have been considered is warranted. It would also be interesting to see how the Prim ordering works in these experiments.

**Questions:**

See weaknesses.

---

### Meta-Review · Area_Chair_mYbE · 2024-12-22

**Metareview:**

**Summary:**
This work focuses on phylogenetic tree inference from a set of genomic sequences. Motivated by issues with the existing methods (low efficiency, low inference accuracy, predetermined node orderings, or high computational cost), the authors introduce a deep learning method MDTree for inferring the topology and branch lengths of the phylogenetic tree from genomic data. Specifically, unlike the existing paradigm that relies on fixed orderings such as lexicographic,  they advocate learning the order in which nodes are added to the phylogenetic tree. The main contribution is extending the existing autoregressive tree (ART) generation method with  an absorbing state diffusion model called the ‘Diffusion Ordering Network’ (DON).  DON sorts the genomic sequences, using embeddings from a pretrained genomic language model, to better capture the similarities between the species from biological and evolutionary perspectives. Furthermore, MDTree can perform parallel node processing.  Empirical validation is conducted on several phylogenetic datasets.

**Strengths:**
Reviewers acknowledged the contributions of this paper, noting the (a) novelty of incorporating dynamic (biological) ordering within the autoregressive setup, (b) efficiency afforded by parallelism, (c) benefits of leveraging pretrained embeddings for phylogenetic inference,  (d) enhanced flexibility over existing ART paradigm, and (e) comprehensive experimental design (encompassing model robustness, runtime, and diversity etc.).


**Weaknesses:**
Some reviewers raised concerns that MDTree seemed to provide rather marginal improvement over ARTree across datasets (despite the very significant computational overhead and increased footprint), so wondered whether dynamic tree ordering was indeed a critical issue for tree reconstruction in practice.
Questions and concerns were also raised about (a) insufficient/at times inaccurate literature review that did not discuss recent related work on Bayesian methods, or conflated different works, (b) lack of motivation for design and complexity of specific architectural choices, (c) missing evaluations for some models, (d) lack of reproducibility of the proposed method, as well as missing information about parameter settings used to obtain results with the baselines, and  (e) issues with presentation that hampered clarity and understanding of the work.

**Recommendation:**
The authors actively participated in the rebuttal phase, satisfactorily addressing some questions and concerns.  However, some concerns remained; e.g., reviewer zB4G maintained that extra computation and carbon footprint did not seem to justify the incommensurately small performance gain. Reviewer i3bg echoed this sentiment, stating they were not convinced about contributions of this work were significant enough.  I fully agree these are  valid concerns. Given the breadth of experiments and the idea of incorporating dynamism and parallelism  in autoregressive generation, I was willing to consider the experimental part to be sufficient.

Reviewers 5UDw and DTGp remained unconvinced about the clarity and presentation, which they found to be confusing. 5UDw maintained that in the current form the work was too tailored to the current setting, and expressed concern that it might not be of interest to the broader ICLR community. DTGp also emphasized several issues with exposition - in particular about the description of DON.

In order to be able to make an informed recommendation, I proceeded to take a closer look at the revised manuscript myself. Despite being (very) familiar with several technical components of this work (such as GCN and diffusion), I found the writing (especially about the technical parts) to be extremely confusing and rather underwhelming. I’m afraid the work in its current form suffers from serious readability issues (as pointed out by 5UDw and DTGp), and a significant effort in form of a major revision (that includes clear mathematical descriptions) needs to be invested before it is ready for publication.

Technical presentation issues aside, I think the paper will significantly benefit from separating the description of key methodological contributions (i.e., incorporating dynamic ordering and parallelism) - which should be clearly developed formally via a mathematical formulation - from the phylogenetic-specific details. Not only will doing so significantly enhance the readability of the paper (and increase confidence about the technical machinery being correct), I believe it will also make the work more broadly accessible.This way the authors could position their methodological contributions  better in the context of previous work on ordering-related issues for autoregressive models (see, e.g., Xu et al. Anytime sampling for Autoregressive Models via Ordered Autoencoding. ICLR 2021),  while still being able to demonstrate the benefits of their overall approach for phylogenetic inference.

**Additional Comments On Reviewer Discussion:**

Details already provided above.

---

### Decision · Program_Chairs · 2025-01-22

Reject